# Short sleep duration is a significant risk factor of obesity: A multicenter observational study of healthy adults in Japan

Mami Takahashi[1,2], Takeshi Shimamoto[1,3], Lumine Matsumoto[1], Yusuke Mitsui[1], Yukari Masuda[1], Hirotaka Matsuzaki[1], Eriko Hasumi[1], Chie Bujo[1], Keiko Niimi[1], Takako Nishikawa[1], Ryoichi Wada[3], Nobutake Yamamichi[1]*

**1** Center for Epidemiology and Preventive Medicine, The University of Tokyo Hospital, Tokyo, Japan, **2** Department of Nursing, The University of Tokyo Hospital, Tokyo, Japan, **3** Kameda Medical Center Makuhari, Chiba, Japan

* NobutakeYamamichi@gmail.com

## Abstract

This multicenter study aimed to elucidate the association between sleep duration and various lifestyle-related disorders in healthy adults in Japan. A total of 62,056 healthy participants (age: 49.4 ± 10.9 years) who received medical checkups from 2010 to 2020 were analyzed cross-sectionally and longitudinally. The mean sleep duration was 6.2 ± 1.0 h in men and 6.1 ± 1.0 h in women. The distribution of sleep duration showed that older people tended to sleep longer, which was clearly observed in men but not in women. Univariate analyses showed that older age, lower body mass index (BMI), habitual drinking, and habitual exercise were significantly associated with longer sleep duration. Multivariate analyses in men showed that sleep duration was positively associated with age, habitual exercise, serum triglyceride (TG), systolic blood pressure (SBP), and habitual drinking and negatively associated with BMI and hemoglobin A1c (HbA1c). Alternatively, in women, sleep duration was positively associated with habitual exercise and TG and negatively associated with BMI, high-density lipoprotein-cholesterol, HbA1c, and current smoking. During the follow-up period, 3,360 of 31,004 individuals (10.8%) developed obesity. The Cox proportional hazards model showed that shorter sleep duration was a significantly higher risk of obesity, and longer sleep duration might be a lower risk of obesity. On the other hand, 1,732 of 39,048 participants (4.4%) developed impaired glucose tolerance, and 6,405 of 33,537 participants (19.1%) developed hypertriglyceridemia. However, the Cox proportional hazards model did not show significant association between sleep duration and impaired glucose tolerance or hypertriglyceridemia. In conclusion, our large-scale cross-sectional study showed that sleep duration was positively associated with habitual exercise and TG and negatively associated with BMI and HbA1c, regardless of sex. Longitudinal analysis revealed that shorter sleep duration is a significant risk factor for obesity.

**Data availability statement:** The data underlying the results presented in the study are available from the corresponding author of the manuscript (nobutakeyamamichi@gmail.com, a private email of corresponding author). In addition, the data are also available from the Center for Epidemiology and Preventive Medicine, The University of Tokyo Hospital via e-mail (dock@h.u-tokyo.ac.jp, the non-author, official institutional point of contact). Neither the ethics committee nor other third-party organizations share our data. We cannot completely open our raw data due to the two reasons below. One reason for the restriction of data is confidentiality of study participants. Another more essential reason is that it is not described in our original study protocol and therefore not approved by the ethic committee of the two medical institutions (University of Tokyo Hospital and Kameda Medical Center).

**Funding:** This study was supported by Grant-in-Aid for Early-Career Scientists of Japan Society for the Promotion of Science (no. 20K19203), Grant-in-Aid for Scientific Research (C) of Japan Society for the Promotion of Science (no. 22K11255), and Grant-in-Aid for Challenging Exploratory Research of Japan Society for the Promotion of Science (no. 22K19665). The funders had no role in study design, data collection and analysis, decision to publish, or preparation of the manuscript.

**Competing interests:** The authors have declared that no competing interests exist.

## Introduction

The relationship between sleep disorders, mortality rates, and various systemic diseases has been currently gaining attention. Since the report of Wingard et al. [1] on the association between sleep patterns and mortality risk, many epidemiological studies showed that short sleep duration is significantly associated with various systemic disorders, such as obesity [2–9], diabetes [3,4,10–12], hypertension [4,13–15], dyslipidemia [16,17], cardiovascular diseases [18–20], cerebrovascular diseases [21–23], and dementia [24–26]. Moreover, not only cross-sectional studies but also several intervention trials have been recently conducted in the Western countries [27,28]. Based on epidemiological evidence indicating the adverse influence of sleep deficiency, sleep debt or the risk of short sleep duration has become an important health-related issue. Consensus statements or recommendations from Europe and the United States proposed an appropriate sleep duration for adults of at least 7 h per night, 7–9 h for young people and adults, and 7–8 h for older individuals [29,30].

Although it is widely believed that promoting healthy sleep duration continuously can prevent various lifestyle-related diseases [29–31], data and research in this field are currently inadequate in Asia. Precise data concerning the effect of sleep duration on health problems are important, especially in Asia, because of the short sleep duration in many Asian countries, such as Japan, South Korea, Philippines, Malaysia, India, Taiwan, Vietnam, and Indonesia [32]. An investigation by the Organization for Economic Development and Cooperation (OECD) also showed that sleep duration was shortest in Japan and South Korea (Gender Data Portal 2021; https://www.oecd.org/gender/data). In the context of these circumstances, we used multicenter large-scale data from healthy people in Japan to perform both cross-sectional and longitudinal analyses to evaluate the effects of sleep on health disorders, such as obesity, hypertension, impaired glucose tolerance, dyslipidemia, and hyperuricemia. By analyzing these lifestyle-related disorders along with age, sex, smoking, drinking, and habitual exercise, we aimed to elucidate the risk of sleep deficiency in various lifestyle-related diseases. Considering the differences in genetics and ethnicity between Asian and Western countries, we hope that sleep health guidance from Japan may also provide insights for other Asian countries.

## Methods

### Study participants

In Japan, those with some disease usually receive medical treatment at hospitals or clinics, which was covered by health insurance. On the contrary, other generally healthy people who do not take medications usually go to medical institute for body health checkup. In this study, we analyzed the generally healthy people (without apparent diseases) who underwent a comprehensive health checkup in all the participating medical institutions in Japan.

Among the total 272,318 generally healthy individuals who underwent a comprehensive health checkup from 2010 to 2020, 67,347 who underwent a health checkup for the first time were selected. All the data concerning laboratory measurements or life-style related factors were derived from the participated medical institutes respectively and merged. After excluding 5,291 individuals with missing data for laboratory test items and/or lifestyle-related questions, 62,056 eligible participants were included in the cross-sectional (retrospective) analysis (Fig 1A).

For the longitudinal (prospective) analysis, 23,350 individuals who have never undergone health checkups until 2020 and 3,259 individuals with insufficient data for analyses were excluded from the above-mentioned 67,347 people (Fig 1B). Consequently, 40,738 baseline participants were included in the longitudinal analysis. All the data were assessed for research purpose in March 2021.

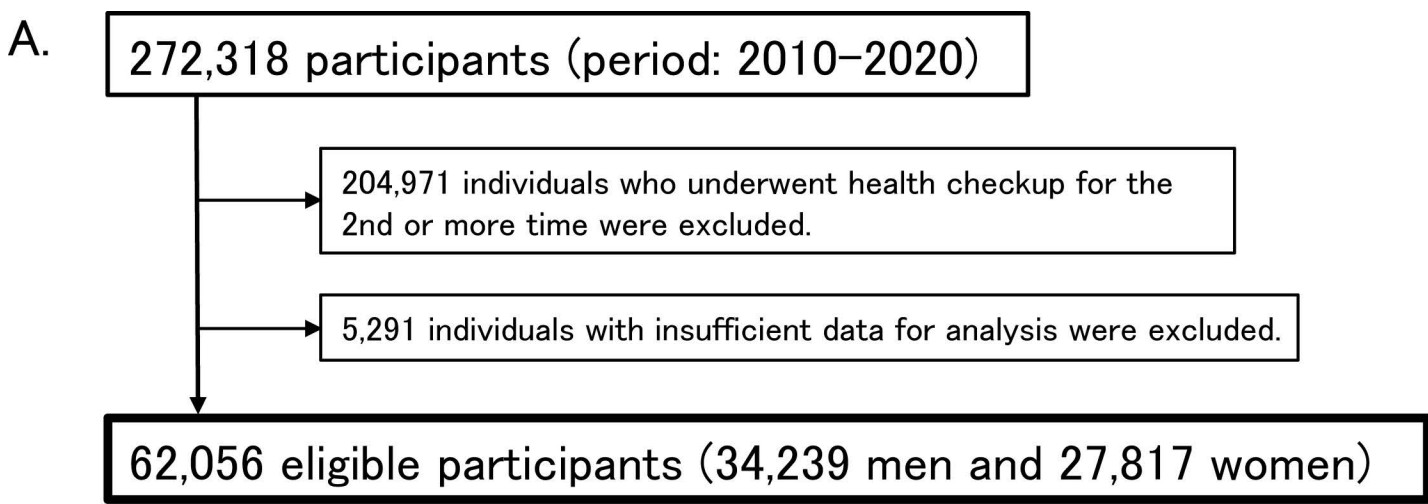

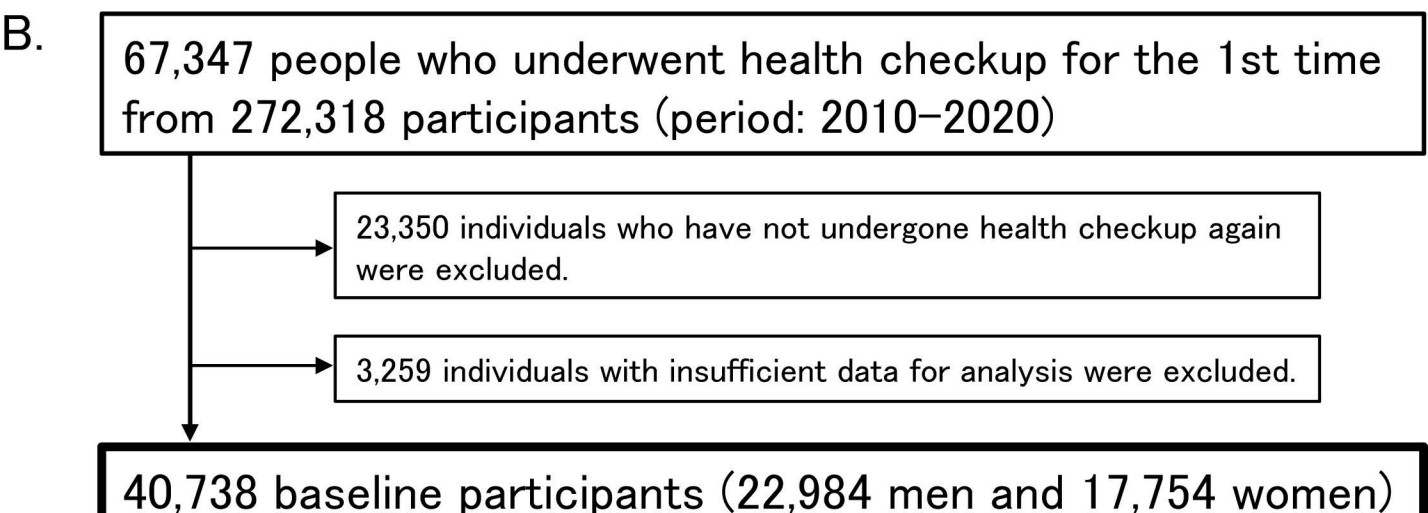

**Fig 1. Study recruitment flowchart.**

### Self-administered questionnaire

A self-administered questionnaire containing questions on sleep duration, smoking habits, alcohol consumption, and habitual exercise was prospectively mailed to the participants a month before the health checkup as a part of our study (from 01/12/2009 to 30/11/2020). Sleep duration was categorized into five groups: < 5, 5–6, 6–7, 7–8, and ≥ 8 h sleep. Other three factors were categorized according to the presence or absence of current smoking, habitual drinking, and habitual exercise.

### Observation items as explanatory or objective variables

The explanatory variables were age, general laboratory test results, and lifestyle factors as follows: 1) age (< 30, 30–39, 40–49, 50–59, 60–69, or ≥ 70 years), 2) body mass index (BMI; < 18.5 kg/m² [underweight], 18.5–25 kg/m² [normal range], or ≥ 25 kg/m² [overweight]),

3) systolic blood pressure (SBP; ≥ 140 mmHg or < 140 mmHg), 4) diastolic blood pressure (DBP; ≥ 90 mmHg or < 90 mmHg), 5) fasting plasma glucose (FPG; ≥ 126 mg/dL or < 126 mg/dL), 6) hemoglobin A1c (HbA1c; ≥ 6.5% or < 6.5%), 7) low-density lipoprotein cholesterol (LDL-C; ≥ 140 mg/dL or < 140 mg/dL), 8) high-density lipoprotein cholesterol (HDL-C; < 40 mg/dL or ≥ 40 mmHg), 9) triglycerides (TG; ≥ 150 mg/dL or < 150 mg/dL), 10) uric acid (UA; > 7.0 mg/dL or ≤ 7.0 mg/dL), 11) current smoking (current smoker or not), 12) habitual drinking ("drinking three or more days per week" or not), and 13) habitual exercise ("at least 30 min at once and at least three times a week" or not). The above 1)-10) factors were quantitative data and shown as mean ± standard deviation (SD) in the table. As for the objective variable, sleep duration was classified into the following five groups: <5 h, 5–6 h, 6–7 h, 7–8 h, or ≥ 8 h.

## Statistical analysis

**Cross-sectional analysis.** The 62,056 study participants were classified into six age groups (<30, 30s, 40s, 50s, 60s, and ≥70s). The distribution of sleeping hours in each age group and comparison of average sleeping hours among the age groups were evaluated using one-way analysis of variance (ANOVA).

Univariate analyses were performed to evaluate whether there were differences in the 13 background factors (age, BMI, SBP, DBP, FPG, HbA1c, LDL-C, HDL-C, TG, UA, smoking, drinking, and habitual exercise) between the five sleep duration groups. ANOVA was applied to the quantitative data, and the $\chi^2$ test was applied to the qualitative data. The Cochran-Armitage test for trends was also used to evaluate the associations between sleep duration and several background factors.

For the multivariate analysis, quantitative variables were normalized, and qualitative variables were changed to 0/1 type dummy variables. Subsequently, after eliminating multicollinearity, multiple regression analysis was conducted, and the standard partial regression coefficient (β) was calculated (sleep duration as the dependent variable and the above-mentioned background factors as independent variables) with a 95% confidence interval (CI).

**Longitudinal analysis.** After excluding those who already had lifestyle-related diseases (obesity, impaired glucose tolerance, or hypertriglyceridemia) at baseline, Cox proportional hazards analyses were performed to evaluate likelihood ratio tests and calculate hazard ratios (HRs) with 95% CI. Based on the results of the cross-sectional analyses and previous studies [33–36], sleep duration, age, BMI, current smoking, habitual drinking, and habitual exercise at baseline were selected as explanatory variables. The references were: 6–7 h for sleep duration, no current smoking, no habitual drinking, no habitual exercise, and BMI < 25.0 kg/m². The object variables were the onsets of three life-related diseases, which were defined when BMI ≥ 25.0 kg/m² for obesity, HbA1c ≥ 6.5% for impaired glucose tolerance, and TG ≥ 150 mg/dL for hypertriglyceridemia during the follow-up period.

All statistical analyses were performed using JMP® 16 software (SAS Institute Inc., Cary, NC, USA) with a significance level of 5%. All analyses were performed separately for males and females because of the significant influence of sex-related genetic background on the onset of lifestyle-related diseases.

**Ethical considerations.** This study was approved by the Ethics Committee of the University of Tokyo School of Medicine and the Clinical Research Review Committee of the University of Tokyo (No. 2498) and Kameda Medical Center (No. 17-075). All methods were carried out in accordance with the Declaration of Helsinki and other relevant guidelines and regulations. All the data were fully anonymized before being accessed by the researchers.

## Results

### Sleep duration of the 62,056 generally healthy participants in Japan

The 62,056 participants comprised 34,239 men and 27,817 women (49.4 ± 10.9 years; range, 18–98 years) (Fig 1A). The mean sleep duration of the study population were 6.2 ± 1.0 h in men and 6.1 ± 1.0 h in women (Table 1). Focusing on the sleep duration, the participants were classified into the following five groups: 2,157 (3.5%) in the < 5 h group, 12,920 (20.8%) in the 5–6 h group, 25,999 (41.9%) in the 6–7 h group, 15,775 (25.4%) in the 7–8 h group, and 5,205 (8.4%) in the ≥ 8 h group. The detailed distributions of sleep duration in men and women are shown in Fig 2 and Table 2. The sleep duration of most participants (87.8% of men and 88.6% of women) ranged from 5 to 8 h.

Fig 2 indicates that older people tend to sleep longer, and an inverse association between sleep duration and age is more apparent in men than in women. Among the five sleep duration groups, the "6–7 h sleep" was the highest in both men (42.2%) and women (41.5%). The proportion of short sleepers (less than 6 h) was higher in women (26.3%) than in men (22.7%). In men, the proportion of short sleepers was the highest in their 30s (26.3%) and 40s (28.5%), whereas it was the highest in their 40s (29.5%) and 50s (30.3%) in women.

### Characteristics of the study participants based on sleep duration and various background factors

The characteristics of the 62,056 participants are presented in Table 2. Univariate analysis showed that most background factors were significantly associated with sleep duration. Sleep duration was significantly associated with age, BMI, SBP, DBP, HbA1c, LDL-C, HDL-C, TG, UA levels, current smoking status, habitual drinking, and habitual exercise in both sexes. Only the FPG level showed a significant association with sleep duration in men, but not in women.

Among the 13 analyzed factors, age, BMI, habitual drinking, and habitual exercise showed clear associations with sleep duration (Table 2). Older men tended to sleep longer, but this relationship was not observed in women. Those with a higher BMI tended to be short sleepers, regardless of sex. For alcohol consumption, both sexes with habitual drinking tended to sleep longer. For habitual exercise, both men and women who were accustomed to physical exercise tended to sleep longer. In contrast, associations between sleep duration and the other nine

**Table 1. Distribution of sleep duration in the 62,056 study participants.**

|  | Sleep duration | | | | | |
|---|---|---|---|---|---|---|
|  | All participants | | Men | | Women | |
| Age groups | N | Mean ± SD | N | Mean ± SD | N | Mean ± SD |
| < 30 | 1,127 | 6.2 ± 1.0 | 696 | 6.1 ± 0.9 | 431 | 6.3 ± 1.1 |
| 30–39 | 11,173 | 6.2 ± 1.0 | 5,828 | 6.1 ± 1.0 | 5,345 | 6.3 ± 1.1 |
| 40–49 | 20,092 | 6.0 ± 1.0 | 10,788 | 6.0 ± 1.0 | 9,304 | 6.1 ± 1.0 |
| 50–59 | 18,058 | 6.1 ± 0.9 | 10,085 | 6.2 ± 1.0 | 7,973 | 6.0 ± 0.9 |
| 60–69 | 9,139 | 6.5 ± 1.0 | 5,441 | 6.6 ± 1.0 | 3,698 | 6.3 ± 1.0 |
| >= 70 | 2,467 | 6.7 ± 1.1 | 1,401 | 6.8 ± 1.1 | 1,066 | 6.5 ± 1.1 |
| Total | 62,056 | 6.2 ± 1.0 | 34,239 | 6.2 ± 1.0 | 27,817 | 6.1 ± 1.0 |
| P-value |  | < 0.0001*** |  | < 0.0001*** |  | < 0.0001*** |

One-way analysis of variance (ANOVA) was used to evaluate the difference in sleep duration between age groups. P-value less than 0.05 was considered statistically significant (***p < 0.001).

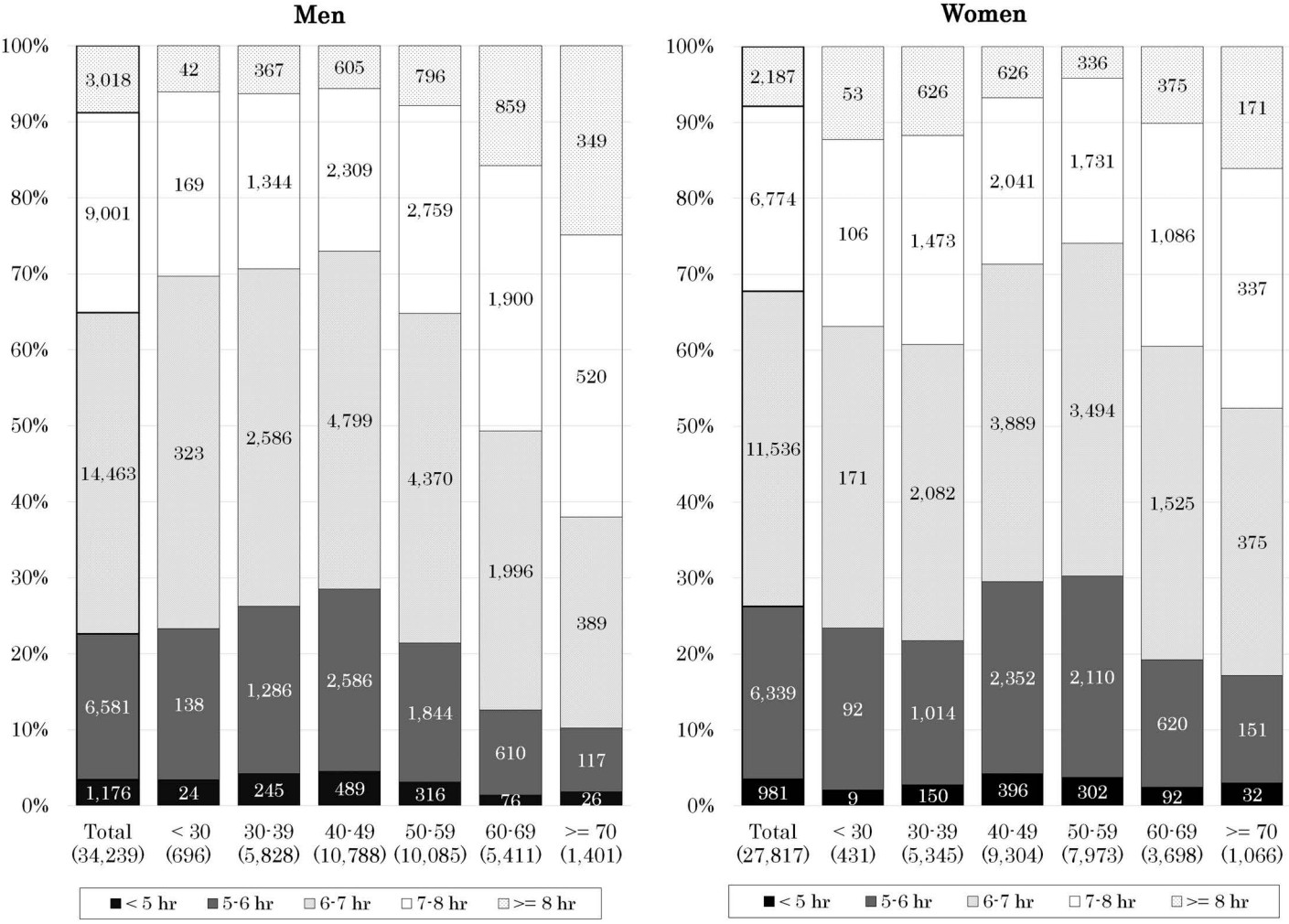

**Fig 2. Distribution of sleep duration by age group.**

factors were not as strong, although most of them were statistically significant in the univariate analysis (Table 2).

Subsequently, the Cochran-Armitage test for trend was performed to evaluate the associations between sleep duration and the 13 background factors. As shown in Fig 3A, all factors other than LDL-C levels were significantly associated with sleep duration in men. However, only 5 of the 13 factors (BMI, HbA1c, current smoking, habitual drinking, and habitual exercise) were significantly associated with sleep duration in women. Fig 3B presents the association of the six background factors with sleep duration. An inverse association between sleep duration and age was clearly observed in men but not in women. Furthermore, we observed an apparent relationship of sleep duration with BMI and habitual exercise, regardless of sex. On the contrary, other factors, such as TG, HbA1c, and HDL-C, did not appear to be linearly associated with sleep duration, although they were statistically significant, at least in men (Fig 3B).

BMI, body mass index; TG, serum triglyceride; HDL-C, high-density lipoprotein cholesterol; HbA1c, hemoglobin A1c; SBP, systolic blood pressure; UA, uric acid; LDL-C, low-density lipoprotein cholesterol; DBP, diastolic blood pressure; FPG, fasting plasma glucose.

**Table 2. Characteristics of the 34,239 male and 27,817 female study participants categorized into five groups according to the length of sleep duration.**

| Variables | Men (n = 34,239) | | | | | | Women (n = 27,817) | | | | | |
|---|---|---|---|---|---|---|---|---|---|---|---|---|
| | Sleep duration groups | | | | | P-value | Sleep duration groups | | | | | P-value |
| | <5hr (n = 1,176) | 5–6hr (n = 6,581) | 6–7hr (n = 14,463) | 7–8hr (n = 9,001) | >= 8hr (n = 3,018) | | <5hr (n = 981) | 5–6hr (n = 6,339) | 6–7hr (n = 11,536) | 7–8hr (n = 6,774) | >= 8hr (n = 2,187) | |
| Age (years) | 46.8±9.5 | 47.3±9.6 | 48.9±10.4 | 51.7±11.5 | 54.9±12.4 | <0.0001*** | 48.8±9.9 | 48.6±9.7 | 49.0±10.5 | 49.3±11.6 | 48.6±13.2 | 0.0014** |
| <30 | 24 (2%) | 138 (2.1%) | 323 (2.2%) | 169 (1.9%) | 42 (1.4%) | <0.0001*** | 9 (0.9%) | 92 (1.5%) | 171 (1.5%) | 106 (1.6%) | 53 (2.4%) | <0.0001*** |
| 30–39 | 245 (20.8%) | 1286 (19.5%) | 2586 (17.9%) | 1344 (14.9%) | 367 (12.2%) | | 150 (15.3%) | 1014 (16%) | 2082 (18%) | 1473 (21.7%) | 626 (28.6%) | |
| 40–49 | 489 (41.6%) | 2586 (39.3%) | 4799 (33.2%) | 2309 (25.7%) | 605 (20%) | | 396 (40.4%) | 2352 (37.1%) | 3889 (33.7%) | 2041 (30.1%) | 626 (28.6%) | |
| 50–59 | 316 (26.9%) | 1844 (28%) | 4370 (30.2%) | 2759 (30.7%) | 796 (26.4%) | | 302 (30.8%) | 2110 (33.3%) | 3494 (30.3%) | 1731 (25.6%) | 336 (15.4%) | |
| 60–69 | 76 (6.5%) | 610 (9.3%) | 1996 (13.8%) | 1900 (21.1%) | 859 (28.5%) | | 92 (9.4%) | 620 (9.8%) | 1525 (13.2%) | 1086 (16%) | 375 (17.1%) | |
| >=70 | 26 (2.2%) | 117 (1.8%) | 389 (2.7%) | 520 (5.8%) | 349 (11.6%) | | 32 (3.3%) | 151 (2.4%) | 375 (3.3%) | 337 (5%) | 171 (7.8%) | |
| BMI (kg/m$^2$) | 24.7±3.8 | 24.2±3.5 | 23.8±3.2 | 23.6±3.1 | 23.7±3.1 | <0.0001*** | 22.6±4.1 | 22.1±3.7 | 21.7±3.4 | 21.4±3.3 | 21.4±3.3 | <0.0001*** |
| <18.5 | 16 (1.4%) | 165 (2.5%) | 340 (2.4%) | 271 (3%) | 87 (2.9%) | <0.0001*** | 120 (12.2%) | 806 (12.7%) | 1574 (13.6%) | 1013 (15%) | 338 (15.5%) | <0.0001*** |
| 18.5–< 25.0 | 686 (58.3%) | 4080 (62%) | 9644 (66.7%) | 6121 (68%) | 2051 (68%) | | 638 (65%) | 4396 (69.3%) | 8330 (72.2%) | 4938 (72.9%) | 1588 (72.6%) | |
| >=25.0 | 474 (40.3%) | 2336 (35.5%) | 4479 (31%) | 2609 (29%) | 880 (29.2%) | | 223 (22.7%) | 1137 (17.9%) | 1632 (14.1%) | 823 (12.1%) | 261 (11.9%) | |
| SBP (mmHg) | 118.8±14.3 | 118.3±14.5 | 118.9±14.7 | 120.1±14.9 | 121.8±15.1 | <0.0001*** | 113.5±16.4 | 113.1±16.3 | 112.8±16.1 | 112.9±16.1 | 111.9±16.0 | 0.0274* |
| SBP>=140 | 90 (7.7%) | 498 (7.6%) | 1178 (8.1%) | 864 (9.6%) | 348 (11.5%) | <0.0001*** | 58 (5.9%) | 350 (5.5%) | 661 (5.7%) | 389 (5.7%) | 117 (5.3%) | 0.9201 |
| DBP (mmHg) | 73.5±11.1 | 73.5±11.1 | 74.0±11.0 | 75.0±10.7 | 75.6±10.7 | <0.0001*** | 70.5±10.6 | 70.4±10.8 | 70.2±10.6 | 70.3±10.4 | 69.6±10.2 | 0.0379* |
| DBP>=90 | 91 (7.7%) | 533 (8.1%) | 1177 (8.1%) | 779 (8.7%) | 279 (9.2%) | 0.1838 | 44 (4.5%) | 291 (4.6%) | 515 (4.5%) | 296 (4.4%) | 87 (4.0%) | 0.8200 |
| FPG (mg/dL) | 101.3±26.9 | 99.4±18.4 | 99.3±17.8 | 100.3±19.1 | 102.3±20.0 | <0.0001*** | 92.3±12.8 | 92.0±15.4 | 91.6±12.8 | 91.4±11.7 | 91.9±13.5 | 0.0586 |
| FPG>=126 | 78 (6.6%) | 355 (5.4%) | 746 (5.2%) | 552 (6.1%) | 242 (8.0%) | <0.0001*** | 23 (2.3%) | 116 (1.8%) | 158 (1.4%) | 99 (1.5%) | 39 (1.8%) | 0.0292* |
| HbA1c (%) | 5.6±0.9 | 5.5±0.7 | 5.5±0.6 | 5.5±0.7 | 5.6±0.7 | <0.0001*** | 5.5±0.6 | 5.5±0.6 | 5.4±0.5 | 5.4±0.4 | 5.4±0.5 | <0.0001*** |
| HbA1c>=6.5 | 88 (7.5%) | 386 (5.9%) | 808 (5.6%) | 577 (6.4%) | 261 (8.6%) | <0.0001*** | 32 (3.3%) | 168 (2.7%) | 253 (2.2%) | 138 (2.0%) | 60 (2.7%) | 0.0199* |
| LDL-C (mg/dL) | 126.7±30.9 | 126.1±30.7 | 126.8±30.3 | 126.5±30.7 | 124.7±31.7 | 0.0145* | 122.0±31.2 | 120.3±32.0 | 120.6±31.6 | 121.3±32.5 | 118.3±32.3 | 0.0016** |
| LDL-C>=140 | 392 (33.3%) | 2087 (31.7%) | 4616 (31.9%) | 2906 (32.3%) | 899 (29.8%) | 0.0925 | 269 (27.4%) | 1633 (25.8%) | 2964 (25.7%) | 1786 (26.4%) | 518 (23.7%) | 0.1050 |
| HDL-C (mg/dL) | 57.8±15.8 | 58.5±14.8 | 59.5±14.9 | 60.0±15.4 | 59.9±16.0 | <0.0001*** | 72.6±17.2 | 73.7±16.4 | 74.3±16.7 | 73.4±16.2 | 72.6±16.9 | <0.0001*** |
| HDL-C < 40 | 88 (7.5%) | 412 (6.3%) | 805 (5.6%) | 479 (5.3%) | 191 (6.3%) | 0.0039** | 12 (1.2%) | 47 (0.7%) | 64 (0.6%) | 34 (0.5%) | 27 (1.2%) | 0.0003*** |
| TG (mg/dL) | 131.2±90.7 | 125.2±93.3 | 125.1±90.8 | 128.2±94.2 | 133.9±100.5 | <0.0001*** | 86.2±56.9 | 79.7±47.2 | 79.7±44.9 | 82.3±52.4 | 83.5±55.4 | < 0.0001*** |
| TG>= 150 | 331 (28.1%) | 1697 (25.8%) | 3640 (25.2%) | 2420 (26.9%) | 882 (29.2%) | <0.0001*** | 99 (10.1%) | 432 (6.8%) | 744 (6.4%) | 500 (7.4%) | 168 (7.7%) | 0.0001*** |
| UA (mg/dL) | 6.2±1.2 | 6.1±1.2 | 6.1±1.2 | 6.1±1.2 | 6.0±1.2 | <0.0001*** | 4.6±1.0 | 4.5±1.0 | 4.4±1.0 | 4.4±1.0 | 4.4±1.0 | <0.0001*** |
| UA > 7.0 | 266 (22.6%) | 1428 (21.7%) | 2917 (20.2%) | 1877 (20.9%) | 595 (19.7%) | 0.0261* | 17 (1.7%) | 83 (1.3%) | 125 (1.1%) | 76 (1.1%) | 36 (1.6%) | 0.0839 |
| Current smoking | 364 (31.0%) | 1880 (28.6%) | 3791 (26.2%) | 2288 (25.4%) | 871 (28.9%) | <0.0001*** | 88 (9.0%) | 468 (7.4%) | 756 (6.6%) | 387 (5.7%) | 175 (8.0%) | <0.0001*** |
| Habitual drinking | 688 (58.5%) | 4138 (62.9%) | 9553 (66.1%) | 6275 (69.7%) | 2123 (70.3%) | <0.0001*** | 316 (32.2%) | 2284 (36.0%) | 4354 (37.7%) | 2506 (37.0%) | 844 (38.6%) | 0.0017** |
| Habitual exercise | 206 (17.5%) | 1468 (22.3%) | 3860 (26.7%) | 2824 (31.4%) | 1073 (35.6%) | <0.0001*** | 162 (16.5%) | 1060 (16.7%) | 2328 (20.2%) | 1566 (23.1%) | 496 (22.7%) | <0.0001*** |

Data show the mean (with SD) or headcount of each variable. Consecutive scale was based on one-way analysis of variance, and the nominal scale was based on the $\chi^2$ test. P-value less than 0.05 was considered statistically significant (*p < 0.05, **p < 0.01, ***p < 0.001). BMI, body mass index; SBP, systolic blood pressure; DBP, diastolic blood pressure; FPG, fasting plasma glucose; HbA1c, hemoglobin A1c; LDL-C, low-density lipoprotein cholesterol; HDL-C, high-density lipoprotein cholesterol; TG, serum triglyceride; UA, uric acid.

*a.*

| Factors | Men | Women | Factors (serum) | Men | Women |
|---|---|---|---|---|---|
| Age (years) | < 0.0001*** | 0.2597 | FPG (mg/dL) | < 0.0001*** | 0.0681 |
| BMI (kg/m²) | < 0.0001*** | < 0.0001*** | HbA1c (%) | < 0.0001*** | 0.0462* |
| SBP (mmHg) | < 0.0001*** | 0.4547 | LDL-C (mg/dL) | 0.0905 | 0.1178 |
| DBP (mmHg) | 0.0108* | 0.1421 | HDL-C (mg/dL) | 0.0275* | 0.4437 |
| Current smoking | 0.0028** | 0.0068** | TG (mg/dL) | 0.0018** | 0.4304 |
| Habitual drinking | < 0.0001*** | 0.0019** | UA (mg/dL) | 0.0148* | 0.4208 |
| Habitual exercise | < 0.0001*** | < 0.0001*** | | | |

*b.*

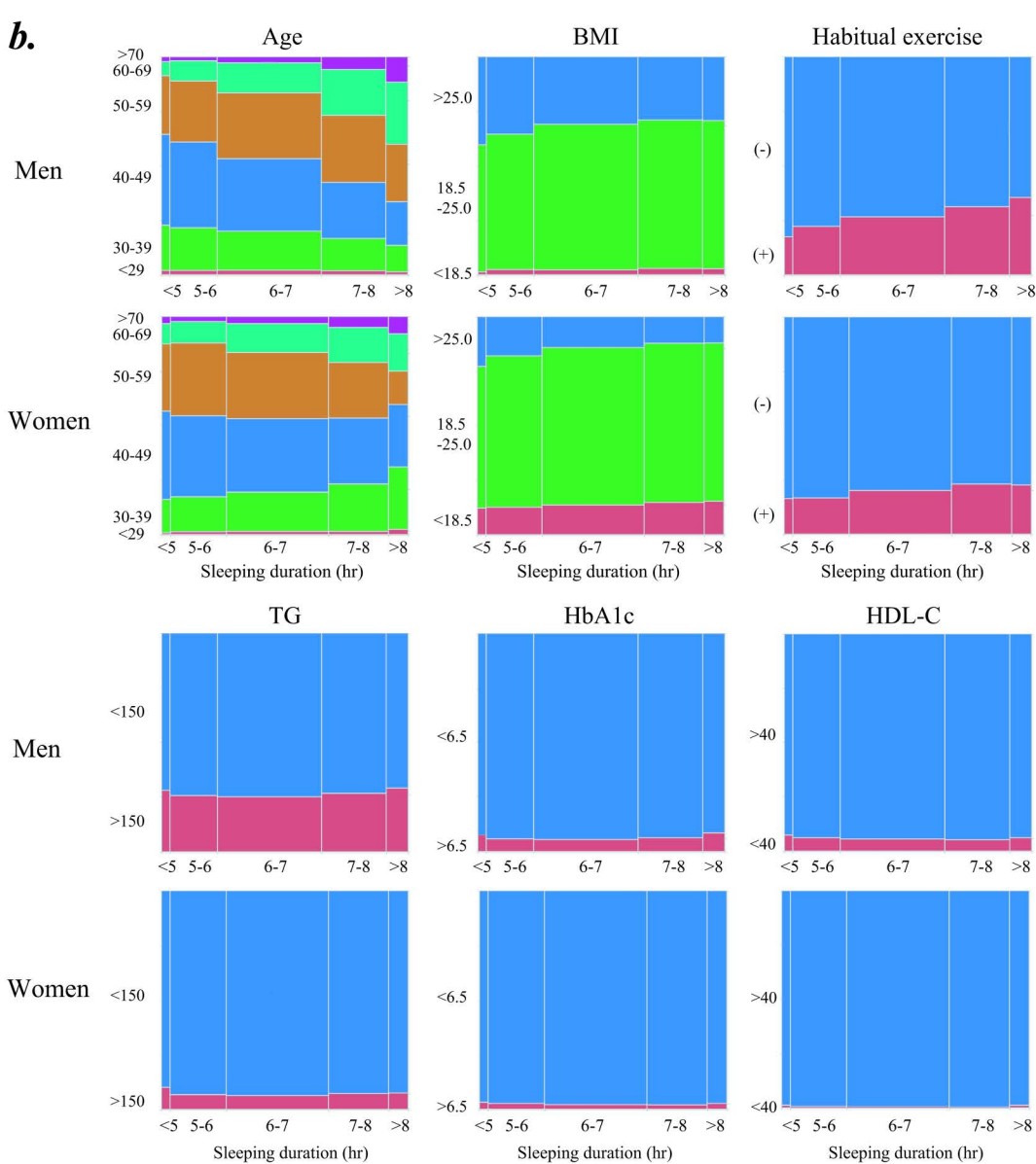

**Fig 3. Relationship between the five groups of sleep duration and background factors analyzed by Cochran-Armitage trend test.** a) All the p values for one-tailed test of 13 factors calculated by Cochran-Armitage trend test (*p < 0.05, **p < 0.01, ***p < 0.001). b) Graphic demonstration of the relation between sleep duration and six background factors (age, BMI, habitual exercise, TG, HbA1c, and HDL-C).

## Multivariate analysis

After considering multicollinearity, two variables (DBP and FPG) were omitted, and multiple regression analysis was performed (Table 3). In men, age, habitual exercise, TG level, SBP, and habitual drinking were significantly associated with longer sleep duration. By contrast, BMI and HbA1c levels were negatively associated with sleep duration. In women, habitual exercise and TG level were significantly associated with longer sleep duration. Whereas, BMI, HDL-C and HbA1c levels, and current smoking status were negatively associated with sleep duration.

Regarding age, a remarkable difference was observed between the sexes: age had the strongest association with sleep duration in men, whereas such an association was not observed in women. For BMI, Table 3 as well as Table 2 shows that short sleep duration was significantly associated with higher BMI regardless of sex, indicating that short sleep duration is a substantial risk factor for obesity. By contrast, habitual exercise was significantly associated with longer sleep duration in both men and women, indicating that appropriate physical exercise may exert a favorable effect on sleep.

Regarding lipid-related blood tests, higher TG (in both men and women) and lower HDL-C (in women only) levels were significantly associated with longer sleep duration (Table 3), suggesting that impaired lipid metabolism may be related to sleep deficits. SBP in men, non-current smoking in women, and habitual drinking in men also showed a significant but weak association with longer sleep duration. No significant associations with sleep duration were observed for UA and LDL-C levels (Table 3).

## Longitudinal analysis

The 40,738 study participants who were selected for the longitudinal analysis (Figure 1B) consisted of 22,984 men (56.4%) and 17,754 women (43.6%), with a mean age of 49.2 ± 10.2 years and a mean sleep duration of 6.2 ± 1.0 h. The total observation period was 11 years (from 2010 to 2020). The mean and median of the observation period are 2,005.3 ± 1,153.6 days and 1,891

**Table 3.  Relationship between sleep duration and background factors according to the multiple regression analysis.**

| Factors | Men (n = 34,239) | | P-value | Women (n = 27,817) | | P-value |
|---|---|---|---|---|---|---|
| | standard partial regression coefficient (b) | 95%C.I. | | standard partial regression coefficient (b) | 95%C.I. | |
| Age | 0.1826 | [0.1715, 0.1936] | <0.0001*** | 0.0123 | [−0.0021, 0.0268] | 0.0942 |
| BMI | −0.1025 | [−0.1154, −0.0896] | <0.0001*** | −0.1022 | [−0.1168, −0.0875] | <0.0001*** |
| Habitual exercise | 0.0631 | [0.0531, 0.0731] | <0.0001*** | 0.0546 | [0.0417, 0.0675] | <0.0001*** |
| TG | 0.0320 | [0.0221, 0.0419] | <0.0001*** | 0.0539 | [0.0308, 0.0769] | <0.0001*** |
| HDL-C | −0.0079 | [−0.0217, 0.0059] | 0.2626 | −0.0448 | [−0.0590, −0.0306] | <0.0001*** |
| HbA1c | −0.0163 | [−0.0261, −0.0064] | 0.0012** | −0.0260 | [−0.0421, −0.0099] | 0.0015** |
| SBP | 0.0427 | [0.0309, 0.0546] | <0.0001*** | 0.0107 | [−0.0026, 0.0241] | 0.1155 |
| Current smoking | 0.0063 | [0.0028, 0.0154] | 0.1758 | −0.0310 | [−0.0492, −0.0128] | 0.0009*** |
| Habitual drinking | 0.0251 | [0.0139, 0.0363] | <0.0001*** | 0.0098 | [−0.0027, 0.0223] | 0.1253 |
| UA | −0.0045 | [−0.0169, 0.0079] | 0.4762 | −0.0078 | [−0.0258, 0.0103] | 0.3995 |
| LDL-C | −0.0033 | [−0.0141, 0.0075] | 0.5484 | −0.0041 | [−0.0172, 0.0090] | 0.5380 |

After excluding DBP and FPG due to multicollinearity, multivariate correlations between sleep duration and each background factor were evaluated. Examined factors were arranged in order of their absolute value of standard partial regression coefficients. P-value less than 0.05 was considered statistically significant (**p < 0.01, ***p < 0.001). BMI, body mass index; TG, serum triglyceride; HDL-C, high-density lipoprotein cholesterol; HbA1c, hemoglobin A1c; SBP, systolic blood pressure; UA, uric acid; LDL-C, low-density lipoprotein cholesterol; DBP, diastolic blood pressure; FPG, fasting plasma glucose.

days respectively. Among the 40,738 participants at baseline, 9,734 showed abnormally high BMI ($\geq$ 25.0 kg/m$^2$), 1,690 showed abnormally high HbA1c levels ($\geq$ 6.5%), and 7,201 showed abnormally high TG levels ($\geq$ 150 mg/dL). Therefore, we performed longitudinal analyses of the remaining 31,004 participants for the development of obesity, 39,048 participants for the development of impaired glucose tolerance, and 33,537 participants for the development of hypertriglyceridemia.

During the follow-up period, 3,360 individuals (10.8%) developed obesity (BMI $\geq$ 25.0). After adjusting for age, smoking, drinking, and habitual exercise, hazard ratios (HR) were calculated using the Cox proportional hazards model. For the five sleep duration groups, the 6–7 h sleep duration group was used as a reference. As shown in Table 4, the shorter sleep duration group had a significantly higher risk of obesity (HR = 1.53 for men and HR = 1.63 for women in the < 5 h group; HR = 1.17 for men and HR = 1.24 for women in the 5–6 h group). Moreover, a longer sleep duration resulted in a lower risk of obesity (HR = 0.86 for men and HR = 0.95 for women in the 7–8 h group; HR = 0.89 for men and HR = 0.98 for women in the $\geq$ 8 h group). In addition to shorter sleep duration, current smoking (HR = 1.33 for men and HR = 1.35 for women) showed a significantly higher risk of obesity. In contrast, habitual drinking (HR = 0.83 for men and HR = 0.87 for women) and habitual exercise in women (HR = 0.80) were associated with a significantly lower risk of obesity.

During the follow-up period, 1,732 participants (4.4%) developed impaired glucose tolerance (HbA1c $\geq$ 6.5%). After adjusting for age, BMI, smoking, drinking, and habitual exercise, HRs were calculated in the same manner. As shown in Table 4, any significant association between sleep duration and impaired glucose tolerance was not detected. For other factors, higher BMI (HR = 3.46 for men and HR = 5.07 for women), current smoking (HR = 1.54 for men and HR = 1.59 for women), and age (HR = 1.05 for men and HR = 1.06 for women) showed a significantly higher risk of impaired glucose tolerance. By contrast, habitual drinking (HR = 0.81 for men and HR = 0.66 for women) showed a significantly lower risk of impaired glucose tolerance.

During the follow-up period, 6,405 participants (19.1%) developed hypertriglyceridemia (TG $\geq$ 150 mg/dL). After adjusting for age, BMI, smoking, drinking, and habitual exercise, HRs were calculated in the same manner. As shown in Table 4, any significant association between sleep duration and hypertriglyceridemia was not found. Regarding other factors, higher BMI (HR = 1.68 for men and HR = 2.76 for women), current smoking (HR = 1.43 for men and HR = 1.92 for women), habitual drinking (HR = 1.15 for men only), and age (HR = 1.03 for women only) were associated with a significantly higher risk of hypertriglyceridemia. In contrast, habitual exercise (HR = 0.90 for men and HR = 0.85 for women) showed a significantly lower risk of hypertriglyceridemia.

## Discussion

In this multicenter study of healthy adults in Japan, we investigated the current trend of sleep duration and analyzed the involvement of sleep duration in various lifestyle-related diseases with multiple background factors. In our cohort, the mean sleep duration was 6.2 ± 1.0 h in men and 6.1 ± 1.0 h in women; the rate of short sleepers was significantly higher in women than in men; and sleep duration was inversely associated with age. In addition, multivariate cross-sectional analyses showed that sleep duration was positively associated with habitual exercise and TG, and negatively associated with BMI and HbA1c in both sexes. Longitudinal analyses revealed that shorter sleep duration was a significant risk factor for obesity.

Our results concerning sleep were consistent with the national big data from Japan (National Health and Nutrition Examination Survey in 2019 (https://www.cdc.gov/nchs/nhanes/index.htm) and Japan Collaborative Cohort Study (JACC Study) [37]). Most people

**Table 4. Multivariate analyses to evaluate the association of sleep duration and other several factors with an onset of obesity, impaired glucose tolerance, or hypertriglyceridemia.**

| Factors | | Men | | | Women | | |
|---|---|---|---|---|---|---|---|
| | | HR | 95%C.I. | P-value | HR | 95%C.I. | P-value |
| | | **Obesity** | | | | | |
| Onset / Non-onset (n = 31,004) | | 2,140 / 13,676 | | | 1,220 / 13,968 | | |
| Sleep duration groups | <5 hr | 1.53 | [1.23, 1.91] | 0.0003*** | 1.63 | [1.23, 2.15] | 0.0013** |
| | 5–6 hr | 1.17 | [1.04, 1.31] | 0.0082** | 1.24 | [1.08, 1.42] | 0.0028** |
| | 6–7 hr | Reference | | | Reference | | |
| | 7–8 hr | 0.86 | [0.78, 0.96] | 0.0082** | 0.95 | [0.82, 1.10] | 0.4937 |
| | >= 8 hr | 0.89 | [0.75, 1.06] | 0.1803 | 0.98 | [0.77, 1.25] | 0.8770 |
| Current smoking | | 1.33 | [1.21, 1.46] | <0.0001*** | 1.35 | [1.09, 1.66] | 0.0072** |
| Habitual drinking | | 0.83 | [0.76, 0.91] | <0.0001*** | 0.87 | [0.77, 0.98] | 0.0168 * |
| Habitual exercise | | 1.05 | [0.96, 1.16] | 0.2952 | 0.8 | [0.69, 0.93] | 0.0039** |
| Age, years | | 1.00 | [0.99, 1.00] | 0.7543 | 1.00 | [0.99, 1.01] | 0.9043 |
| Model χ2 test | | p < 0.0001*** | | | p < 0.0001*** | | |
| | | **Impaired Glucose Tolerance** | | | | | |
| Onset / Non-onset (n = 39,048) | | 1,276 / 20,383 | | | 456 / 16,933 | | |
| Sleep duration groups | <5 hr | 1.21 | [0.90, 1.64] | 0.2244 | 1.46 | [0.96, 2.22] | 0.0894 |
| | 5–6 hr | 1.16 | [0.99, 1.34] | 0.0613 | 0.97 | [0.77, 1.23] | 0.8135 |
| | 6–7 hr | Reference | | | Reference | | |
| | 7–8 hr | 0.91 | [0.79, 1.04] | 0.1785 | 0.98 | [0.77, 1.25] | 0.8847 |
| | >= 8 hr | 0.91 | [0.75, 1.12] | 0.3832 | 0.70 | [0.45, 1.10] | 0.1040 |
| BMI >= 25 kg/m$^2$ | | 3.46 | [3.10, 3.87] | <0.0001*** | 5.07 | [4.21, 6.11] | <0.0001*** |
| Current smoking | | 1.54 | [1.37, 1.73] | <0.0001*** | 1.59 | [1.12, 2.26] | 0.0158 * |
| Habitual drinking | | 0.81 | [0.72, 0.91] | 0.0005*** | 0.66 | [0.53, 0.81] | <0.0001*** |
| Habitual exercise | | 0.97 | [0.85, 1.10] | 0.6118 | 1.04 | [0.83, 1.30] | 0.7286 |
| Age, years | | 1.05 | [1.05, 1.06] | <0.0001*** | 1.06 | [1.05, 1.07] | <0.0001*** |
| Model χ2 test | | p < 0.0001*** | | | p < 0.0001*** | | |
| | | **Hypertriglyceridemia** | | | | | |
| Onset / Non-onset (n = 33,537) | | 4,491 / 12,421 | | | 1,914 / 14,711 | | |
| Sleep duration groups | <5 hr | 1.04 | [0.88, 1.23] | 0.6277 | 1.14 | [0.91, 1.44] | 0.2686 |
| | 5–6 hr | 0.99 | [0.91, 1.07] | 0.7929 | 1.01 | [0.90, 1.13] | 0.8232 |
| | 6–7 hr | Reference | | | Reference | | |
| | 7–8 hr | 0.96 | [0.89, 1.04] | 0.2985 | 0.95 | [0.84, 1.07] | 0.3879 |
| | >= 8 hr | 1.07 | [0.96, 1.20] | 0.2228 | 1.00 | [0.83, 1.21] | 0.9994 |
| BMI >= 25 kg/m$^2$ | | 1.68 | [1.58, 1.79] | <0.0001*** | 2.76 | [2.49, 3.05] | <0.0001*** |
| Current smoking | | 1.43 | [1.34, 1.53] | <0.0001*** | 1.92 | [1.65, 2.25] | <0.0001*** |
| Habitual drinking | | 1.15 | [1.07, 1.23] | <0.0001*** | 0.92 | [0.84, 1.01] | 0.0898 |
| Habitual exercise | | 0.90 | [0.85, 0.97] | 0.0035** | 0.85 | [0.75, 0.95] | 0.0055** |
| Age, years | | 1.00 | [1.00, 1.00] | 0.9167 | 1.03 | [1.02, 1.03] | <0.0001*** |
| Model χ2 test | | <0.0001*** | | | <0.0001*** | | |

To analyze the risk of obesity, smoking, drinking, exercise, and age were adjusted. To analyze impaired glucose tolerance and hypertriglyceridemia, BMI, smoking, drinking, exercise, and age were adjusted. P-value less than 0.05 was considered statistically significant (*p < 0.05, **p < 0.01, ***p < 0.001). BMI, body mass index; TG, serum triglyceride; HbA1c, hemoglobin A1c; SBP, systolic blood pressure; HR, hazards ratio; CI, confidence interval.

sleep for less than 6–7 h; sleep duration is especially shorter in the working and child-rearing generations, and older people tend to sleep longer. Except for the short sleeping hours in Japan, various sleep characteristics in our cohort were similar to those reported in the United States [38]. Our results also showed that the average sleep duration was significantly shorter in women than in men (Table 1). According to a survey by the OECD (Gender Data Portal 2021; https://www.oecd.org/gender/data), the sleep duration of women was shorter than that of men in only seven (21.2%) of the 33 surveyed countries, including Japan. The OECD survey also reported that the average sleep time of Japanese women was one of the shortest in several countries and 13 min shorter than that of Japanese men. These data are comparable to ours, indicating that our data properly represent the sleep patterns in Japan.

According to previous epidemiological studies in Europe and the United States, a U-shaped or inverse linear relationship was observed between sleep duration and obesity in adults [3,6,39], meaning that "both short and long" or "only short" sleep duration increase the risk of obesity. Short sleep duration is generally considered a significant risk factor for obesity in the Western countries based on both cross-sectional and longitudinal analyses [8,40,41]. Similarly, our large-scale analyses showed that a short sleep duration of less than 7 h is a significant risk factor for the development of obesity. An association between short sleep duration and obesity has been confirmed by the results of recent intervention studies on sleeping hours [27,28] and can also be partly explained by some physiological mechanisms [42,43]. About the physiological mechanisms, it was reported that short sleep duration was associated with reduction of anorexigenic hormone leptin and elevations of orexigenic factor ghrelin, which can lead to increased hunger and appetite [42].

Demonstrating the significant association between short sleep duration and obesity in Asia is one of the main findings of our study. Our results are reliable and valuable such that they were derived from large-scale data of generally healthy people in Japan, and that both cross-sectional and longitudinal analyses yielded the same results.

Sex differences in the effects of short sleep duration on obesity remain unclear. Some reports have shown the effect of short sleep duration on obesity in both sexes [8,44,45], whereas others have shown it only in men [46–50] or women [41,51]. Our current study revealed a significant association between short sleep duration and obesity in both sexes, which is similar to the report of Xiao et al. [45] from the United States. Despite the low prevalence of obesity in Japan, our study showed the effect of short sleep duration on obesity regardless of sex, probably due to the large number of study participants. Similar to the largest longitudinal study of women in the United States [52], we believe that short sleep duration is a risk factor for obesity in not only men but also women. However, sleep-related studies from Asia remain inadequate presently; more analyses are necessary in the future in terms of race, age, sex, study design, and confounding factors (demographic, socioeconomic, lifestyle-related, etc.).

Regarding lifestyle-related diseases other than obesity, our cross-sectional study showed that short sleep duration was negatively associated with HbA1c levels and positively associated with TG levels. The involvement of short sleep duration in the onset of diabetes has been reported not only in the Western countries [3,4,10,11,44,53] but also in Japan [54,55], indicating that our results are valid. On the contrary, the increase in TG levels with prolonged sleep remains controversial, although some reports have shown an association between TG levels and sleep duration [56,57]. Apart from obesity, our longitudinal analyses did not show a significant association between sleep duration and impaired glucose tolerance or hypertriglyceridemia. To resolve these issues, more large-scale prospective studies in Asia are necessary in the future.

In summary, our current study, based on the data of generally healthy people in Japan, showed that short sleep duration was significantly associated with lifestyle-related diseases, such as obesity, impaired glucose tolerance, and hypertriglyceridemia. It is especially

noteworthy that cross-sectional multivariate and longitudinal analyses using a Cox proportional hazard model both showed that a short sleep duration had a significant negative association with BMI, regardless of sex, indicating that sleep deficiency is a definitive risk factor for obesity. Based on these results, we plan to evaluate whether sleep education is useful for preventing lifestyle-related diseases, including obesity. In contrast to short sleep duration, the influence of long sleep duration on lifestyle-related diseases was not clearly observed in our study. Although some reports have suggested that long sleep duration has an unfavorable effect on health [11], no consensus has been reached worldwide.

In addition to several lifestyle-related disorders, our multiple regression analysis (Table 3) showed a significant association between sleep duration and habitual exercise regardless of sex. Although the cross-sectional analysis listed in Table 3 did not show causal relationships, our results probably reflect the beneficial effects of exercise on sleep-related items, including sleep duration, as indicated by several previous studies [33,58–60]. Our results suggest that regular exercise can effectively improve sleep-related habits.

This study has some limitations. First, because of the cross-sectional or prospective observational design of the study, we were unable to perform accurate analyses of causal effects. Second, we could not completely exclude participants with some latent disease or health problems, including obstructive sleep apnea and narcolepsy, sleep modifier medications, thyroid dysfunctions, menopausal symptoms, various chronic diseases (hypertension, diabetes, dyslipidemia, etc.), and so on. However, the percentage of such people who had some health disorders was quite small, because in principle, the health checkup programs were prepared and provided with generally healthy people. Third, the influence of other confounding factors on the relationship between sleep duration and various explanatory variables, such as income, occupation, diet, shift work, etc. remains possible. Fourth, sleep duration was self-reported in the questionnaire. Measuring sleep duration using sleep-monitoring devices may be needed to evaluate sleep duration more objectively. Fifth, this study lacks data on ethnic groups within the population. However, in the health checkup in Japan, almost all of them have Japanese nationality. As a result, the results from our study certainly reflect the data of sleep in Japan and therefore in East Asia.

## Conclusions

Our large-scale cross-sectional study showed that sleep duration was negatively associated with BMI and HbA1c and positively associated with habitual exercise and serum TG levels, regardless of sex. Longitudinal analyses revealed that short sleep duration is a significant risk factor for obesity.

## Acknowledgments

The authors thank Editage (www.editage.com) for English language editing.

## Author contributions

**Conceptualization:** Mami Takahashi.

**Data curation:** Mami Takahashi, Takeshi Shimamoto.

**Formal analysis:** Mami Takahashi, Takeshi Shimamoto.

**Project administration:** Nobutake Yamamichi.

**Writing – original draft:** Mami Takahashi.

**Writing – review & editing:** Takeshi Shimamoto, Lumine Matsumoto, Yusuke Mitsui, Yukari Masuda, Hirotaka Matsuzaki, Eriko Hasumi, Chie Bujo, Keiko Niimi, Takako Nishikawa, Ryoichi Wada, Nobutake Yamamichi.

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
