## [Decision Letter · Decision Letter 0]

9 Oct 2024

PONE-D-24-37623Short sleep duration is a significant risk factor of obesity: A multicenter observational study of healthy adults in JapanPLOS ONE

Dear Dr. Yamamichi,

Thank you for submitting your manuscript to PLOS ONE. After careful consideration, we feel that it has merit but does not fully meet PLOS ONE’s publication criteria as it currently stands. Therefore, we invite you to submit a revised version of the manuscript that addresses the points raised during the review process.

We look forward to receiving your revised manuscript.

Kind regards,

Hidetaka Hamasaki

Academic Editor

PLOS ONE

Journal Requirements:

 “This study was supported by Grant-in-Aid for Early-Career Scientists (no. 20K19203), Grant-in-Aid for Scientific Research (C) (no. 22K11255), and Grant-in-Aid for Challenging Exploratory Research (no. 22K19665).”

4. In this instance it seems there may be acceptable restrictions in place that prevent the public sharing of your minimal data. However, in line with our goal of ensuring long-term data availability to all interested researchers, PLOS’ Data Policy states that authors cannot be the sole named individuals responsible for ensuring data access (http://journals.plos.org/plosone/s/data-availability#loc-acceptable-data-sharing-methods).

6. Please include captions for your Supporting Information files at the end of your manuscript, and update any in-text citations to match accordingly. Please see our Supporting Information guidelines for more information: http://journals.plos.org/plosone/s/supporting-information .

Reviewers' comments:

Reviewer's Responses to Questions

**Comments to the Author**

1. Is the manuscript technically sound, and do the data support the conclusions?

Reviewer #1: Yes

Reviewer #2: Yes

Reviewer #3: Partly

Reviewer #4: Yes

2. Has the statistical analysis been performed appropriately and rigorously? 

Reviewer #1: Yes

Reviewer #2: Yes

Reviewer #3: Yes

Reviewer #4: Yes

3. Have the authors made all data underlying the findings in their manuscript fully available?

Reviewer #1: No

Reviewer #2: Yes

Reviewer #3: No

Reviewer #4: Yes

4. Is the manuscript presented in an intelligible fashion and written in standard English?

Reviewer #1: Yes

Reviewer #2: Yes

Reviewer #3: No

Reviewer #4: Yes

5. Review Comments to the Author

Reviewer #1: This multicenter study examined the link between sleep duration and lifestyle-related disorders in Japanese population. The results showed longer sleep is linked to better health markers, while shorter sleep increases obesity risk. The findings of the study are valuable for promoting healthy lifestyles among local and Asian populations. However, I have several points of criticism for the authors to consider.

1.The study used healthy adults who came for a checkup. How did authors define a healthy population? How was the unhealthy population excluded? The exclusion criteria were not mentioned in the method section or Figure 1.

2.The study lacks data on ethnic groups within the population. The authors mention that the findings are relevant to an Asian demographic. Yet, the population in this study is likely dominated by East Asians. Please provide basic information on the ratios of other ethnic groups like South Asian, Caucasian, and Black.

3.Follow-up study: How long did the authors monitor the population? This needs clarification. Was the follow-up period sufficient to identify HR, particularly for glucose intolerance? The event rate for glucose intolerance is lower compared to the other two.

4.Quantitative data in Table 2 are not defined as mean ± SD, which should be detailed in the methods section.

Reviewer #2: 1. The manuscript is technically sound, and the data do support the conclusions.

2. The statistical analysis has been performed appropriately and rigorously.

3. The authors have made all data underlying the findings in their manuscript fully available.

4. The manuscript is presented in an intelligible fashion and written in standard English.

Reviewer #3: The study titled ‘short sleep duration is a significant risk factor of obesity: A multicenter observational study of healthy adults in Japan’ aimed to document the associations between sleep duration and the development of some metabolic distubances (obesity, glucose intolerance, dyslipidemia). The authors identified a cohort who had been received medical checkups from 2010 to 2020, they analyzed data of these participants cross-sectionally and longitudinally. After presenting the results, the authors conclude that shorter sleep duration is a significant risk factor for obesity.

The study is potentially attractive with its huge study population and popular topic. On the other hand, the presentation should be improved to clear doubts related to confounding factors.

1.The methodology is not clear. I suggest the authors to provide more detail about the medical check-up, the facility care of the study participants had been evaluated, the laboratory measurements (in a central or in various centers), the inclusion and exclusion criteria in more detail

2.Does Table 2 identify baseline characteristics of the study participants?

3.How many of the study participants were shift-workers?

4.How many of the study participants had thyroid dysfunctions?

5.How many of the study participants had been diagnosed with chronic diseases (hypertension, diabetes, dyslipidemia)?

6.How many of the study participants had been on sleep modifier medications, including antidepresants, antihistaminics, etc?

7.Did the authors have data related to family history of diabetes, dyslipidemia, etc?

8.Did the authors have data on menopause status of women participants? This may have impact on sleep duration

9.How many of the participants had been on hypolipidemic, antihypertensive drugs when evaluated longitudinally.

10.I suggest the authors to include triglyceride to HDL-C ratio as a surrogate marker of insulin resistance and may be considered a reliable marker for such a study.

11.In Table 2, I suggest the authors to provide the percentages of age in parenthesis.

12.In Discussion, it should be emphasized that most of the study population had normal BMI, and the obese patients constituted the minority. Such a population may not reflect the importance of the association between sleep duration and the investigated parameters.

Reviewer #4: The study is interesting and carries an important message on sleep.

The manuscript is clearly written however some minor revision is required

Please define what is habitual drinking and also habitual exercise?

In the discussion, can authors elaborate on the physiological mechanism of sleeping as mentioned.

6. PLOS authors have the option to publish the peer review history of their article (what does this mean? ). If published, this will include your full peer review and any attached files.

**Do you want your identity to be public for this peer review?** For information about this choice, including consent withdrawal, please see our Privacy Policy .

Reviewer #1: No

Reviewer #2: **Yes: ** Kassa Demissie Abdi (PhD)

Reviewer #3: No

Reviewer #4: No

---

## [Author Response · Author response to Decision Letter 1]

7 Jan 2025

All the response are described in the uploaded file "Response to Reviewers 20250104_Yamamichi.docx".

---

## [Decision Letter · Decision Letter 1]

22 Jan 2025

PONE-D-24-37623R1Short sleep duration is a significant risk factor of obesity: A multicenter observational study of healthy adults in JapanPLOS ONE

Dear Dr. Yamamichi,

Thank you for submitting your manuscript to PLOS ONE. After careful consideration, we feel that it has merit but does not fully meet PLOS ONE’s publication criteria as it currently stands. Therefore, we invite you to submit a revised version of the manuscript that addresses the points raised during the review process.

We look forward to receiving your revised manuscript.

Kind regards,

Hidetaka Hamasaki

Academic Editor

PLOS ONE

Reviewers' comments:

Reviewer's Responses to Questions

**Comments to the Author**

1. If the authors have adequately addressed your comments raised in a previous round of review and you feel that this manuscript is now acceptable for publication, you may indicate that here to bypass the “Comments to the Author” section, enter your conflict of interest statement in the “Confidential to Editor” section, and submit your "Accept" recommendation.

Reviewer #1: All comments have been addressed

Reviewer #2: All comments have been addressed

Reviewer #3: All comments have been addressed

2. Is the manuscript technically sound, and do the data support the conclusions?

Reviewer #1: Yes

Reviewer #2: Yes

Reviewer #3: Partly

3. Has the statistical analysis been performed appropriately and rigorously? 

Reviewer #1: Yes

Reviewer #2: Yes

Reviewer #3: Yes

4. Have the authors made all data underlying the findings in their manuscript fully available?

Reviewer #1: No

Reviewer #2: Yes

Reviewer #3: No

5. Is the manuscript presented in an intelligible fashion and written in standard English?

Reviewer #1: Yes

Reviewer #2: Yes

Reviewer #3: Yes

6. Review Comments to the Author

Reviewer #1: I conglutinate authors. All the comments have been addressed. The manuscript is now in a good shape for publication.

Reviewer #2: 1. All comments have been addressed.

2. The manuscript is technically sound, and the data supported the conclusions.

3. The statistical analysis has been performed appropriately and rigorously.

4. The authors have made all data underlying the findings in their manuscript fully available.

5. The manuscript is presented in an intelligible fashion and written in standard English.

Reviewer #3: The authors have revised the raised concerns properly. It has much improved. However, I have some more comments:

1. HDL-C levels differ between sex. Therefore, the cutoff values should be different for men (40 mg/dL) and for women (50 mg/dl).

2. Line 149, no habitual exercise, probably a typo error and it should be ‘habitual exercise’. Please check

3. Since fasting plasma glucose, postprandial plasma glucose, and HOMA-IR were not available, I suggest the authors to provide triglyceride to HDL-C ratio (TG/HDL-C) as a surrogate marker of insulin resistance. Please consider such an analysis.

4. The total observation period was 11 years. But what was the median or mean follow-up duration. Please specify. Were all of the participants with a follow-up duration shorter than a certain period (for example 1-year) excluded? please clarify

5. In regard to sleep duration, was it validated in sequential visits, or just the baseline data were considered in the analysis? Was there much participants with changing sleep habits? Please clarify.

7. PLOS authors have the option to publish the peer review history of their article (what does this mean? ). If published, this will include your full peer review and any attached files.

**Do you want your identity to be public for this peer review?** For information about this choice, including consent withdrawal, please see our Privacy Policy .

Reviewer #1: No

Reviewer #2: **Yes: ** Kassa Demissie Abdi (PhD)

Reviewer #3: No

---

## [Author Response · Author response to Decision Letter 2]

25 Jan 2025

We answered all the comments in the "Response to Reviewers 20250125_Yamamichi.docx" which was uploaded.

---

## [Editor Report · Decision Letter 2]

28 Jan 2025

Short sleep duration is a significant risk factor of obesity: A multicenter observational study of healthy adults in Japan

PONE-D-24-37623R2

Dear Dr. Yamamichi,

We’re pleased to inform you that your manuscript has been judged scientifically suitable for publication and will be formally accepted for publication once it meets all outstanding technical requirements.

Kind regards,

Hidetaka Hamasaki

Academic Editor

PLOS ONE
---

## [Editor Report · Acceptance letter]

PONE-D-24-37623R2

PLOS ONE

Dear Dr. Yamamichi,

I'm pleased to inform you that your manuscript has been deemed suitable for publication in PLOS ONE. Congratulations! Your manuscript is now being handed over to our production team.

Kind regards,

on behalf of

Dr. Hidetaka Hamasaki

Academic Editor

PLOS ONE